# Public conservation connection and support between ocean and terrestrial systems in the United States

**Halley E. Froehlich**[1,2]*, **Darien D. Mizuta**[1¤], **Jono R. Wilson**[3,4]

**1** Department of Ecology, Evolution and Marine Biology, University of California, Santa Barbara, California, United States of America, **2** Environmental Studies, University of California, Santa Barbara, California, United States of America, **3** Bren School of Environmental Science & Management, University of California, Santa Barbara, California, United States of America, **4** Oceans Program, The Nature Conservancy California, Santa Barbara, California, United States of America

¤ Current address: Natural Resources, Virginia Institute of Marine Science, William & Mary, Gloucester Point, Virginia, United States of America

* hefroehlich@ucsb.edu

**Data Availability Statement:** All data and code are publicly available on the Froehlich Lab GitHub repo: https://github.com/Froehlich-Lab/landsea_usa.

**Funding:** This project was supported by funding from the The Nature Conservancy (FH02P). Role of

## Abstract

Terrestrial and ocean ecosystems are increasingly under threat from an array of anthropogenic pressures. And while threats mount, how people view and value nature is changing. In the United States (U.S.) in particular, there is a shift away from viewing nature as something to 'dominate,' as evidenced in the decline in hunting. However, it is unclear if or how opinions around environmental issues and conservation need might differ when comparing ocean versus terrestrial ecosystems, especially given the prevalence and continued importance of wild capture fishing in the U.S. We employed two national parallel surveys, one focused on oceans, the other land, receiving responses from nearly every state in the U.S. (N = 1,973). While we found only slight, but statistically significant more concern for ocean habitats and animals over terrestrial ecosystems, this did not translate to increased willingness to monetarily support more ocean conservation actions. Using Random Forest models, we also found the best predictor of conservation need was feeling most impacted by environmental issues personally (self and/or community), regardless of ecosystem type. In fact, land versus sea (survey) had the lowest rank in the models, underscoring the importance of general nature-based interactions. Instead, the number of outdoor recreational activities was a highly ranked variable explaining the level of reported impact to self/community, with people who participate in 2 or more activities scoring higher levels of impact, on average. Notably, people who hunt *and* fish, versus only do one or the other, reported higher levels of impact and participated in more activities overall, providing a more nuanced finding regarding the nature 'dominance hypothesis.' Voting, not political affiliation, was also important in explaining responses, and governmental mechanisms to fund conservation were favored over voluntary. Overall, our results add to the strong existing literature that access and connection to nature is key, but uniquely broad connection may "float all boats," especially when diversified.

Funder: Dr. Jono Wilson, co-author and employee of The Nature Conservancy, contributed to the idea generation, survey question design, and provided constructive feedback on the first draft of the manuscript.

Competing interests: Jono Wilson is an employee of the Nature Conservancy. This does not alter our adherence to PLOS ONE policies on sharing data and materials.

## Introduction

As land and seas are under mounting anthropogenic pressures–largely from agriculture, development, fishing, pollution, and climate change [1, 2]–there has been a shift in social value over time, i.e. ideals and principles informing human behavior, towards protecting and conserving biodiversity in the United States (U.S.) [3]. While these shifts are likely non-linear and the root of these long-term changes is challenging to pinpoint, there is some evidence such large-scale societal change is generational and tied to commiserate declines in activities of 'dominance' over nature (e.g., hunting), where wildlife is treated as only a resource for humans. However, conservation and attitudes towards such efforts are not homogenous and there may be distinct views on what is most important in predicting social value of conservation or biodiversity, including the type of environment at risk.

Social views and values of conservation may differ between terrestrial and marine ecosystems. Likely due to a combination of relative accessibility and ease of observation, among other factors, most U.S. federal policies for conservation are rooted in land-based protections, while ocean-based efforts–mostly fishery reforms and passive approaches of marine protected areas (MPAs)–are comparatively recent [4, 5]. Matching the terrestrial conservation trend, funding for conservation projects is typically terrestrial dominated [6] and private-land easements, an exponentially growing sector to conserve land in the U.S. [7, 8], are less likely in marine systems, a more common-use space. The collective conservation efforts for the terrestrial ecosystem may lend to more favorable perception of the conditions, thus reduced conservation needs. In comparison, fishing is still a major part of society and source of seafood, including in the U.S. [9, 10], which if the 'dominance hypothesis' holds [3], should result in perceived lower threat and need of intervention. The 'dominance hypothesis' proposes when nature is treated or seen primarily as a resource to exploit for humans there is less consideration for ecosystems. Yet overall, it is unclear whether distinct feelings towards different animals and habitats on land versus sea influence overall views on the need for conservation.

Collective social voice can affect change, but it is likely influenced by personal associations with nature and how those activities can or should be funded. Education or familiarity with conservation (i.e., conservation literacy, such as knowledge of how an ecosystem is impacted) or sustainability activities (e.g., consumer choice) may play a role in the value of conservation action (e.g., [11, 12]). Indeed, personal experiences in general can have an oversized impact on people's concern about the environment [13, 14]. For example, how people engage with nature through various recreational activities (e.g, hunting, fishing, hiking) differ and can influence perceived benefit or value of a place or animal [15–17]. Such activities, especially hunting, fishing, and access to state parks, fund governmental conservation programs, but do not necessarily capture the full dimension of how people view conservation needs (or impacts) and how to financially address those needs [18].

Lastly, polarization is seemingly at an all-time high in the U.S., impacting how people perceive and trust certain fields (e.g., science and education) [19] and related topics (e.g., vaccines and climate change) [20, 21]. However, while there may be strong ideological lines for certain topics across the U.S.–past research documenting an increasing divide around environmental protection spending in the U.S. [22]–views on conservation of biodiversity on land versus the sea have not been explored. Broadly, it might be assumed such a topic falls in line with climate change given its role in biodiversity loss [23] and 'party sorting theory' [22]. Yet, others have found political divides can be shallower than expected, evidenced by bipartisan state climate policies and actions [24, 25], offering hope for collective support for conservation of wildlife and ecosystems into the future.

In this study, we explore how perceptions of conservation may differ in distinct ways concerning land and sea. We specifically compared differences in responses to conservation issues,

need, interventions and monetary support for the oceans versus land using a parallel survey design. In addition, we explored the predictive power of certain attributes (e.g., age, politics, outdoor recreation), that have been found to influence value, on how people across the U.S. report their connectedness to an ecosystem (via impact of environmental issue to self or community) and overall need for conservation intervention. We paid particularly close attention to those who hunt and/or fish.

## Methods

We developed two parallel surveys, with a mix of binary, rank/likert, and open-ended questions, one focused on oceans and one focused on land. Both surveys contained 21 total questions ranging from general characteristics of the respondent, including age, education, gender, and political affiliation, to involvement and viewpoints pertaining to conservation concerns, interventions, and funding. Questions were informed by the standing literature and developed in consultation with key terrestrial and ocean experts from The Nature Conservancy, as well as specialized survey designers to ensure the questions achieved the intended goal. Employing two separate surveys focused on different environments allowed us to explicitly test the strength of conservation views around ocean versus terrestrial conservation. To view the non-demographic survey questions used in the analyses, see (S1 Table in S1 File).

The surveys were administered by PollFish, August 11–17, 2021, an online survey tool where respondents are invited using a double opt-in, meaning interest to participate in any surveys is determined, regardless of what survey, unique user IDs are created, and they join the potential respondent pool if they fit the target audience. In our case, the respondents had to be 18+ years old and live in the U.S. The survey was University of California, Santa Barbara IRB approved on 6/1/2021 (protocol 3-21-0394). Out of an attempted 2000 respondents, a total of 1,973 were viable.

### Descriptive difference is ocean vs terrestrial reponses

Several questions were specifically designed to get at the difference in perceived conservation need and threat to the given ecosystems. Out of the 21 questions, we individually evaluated eight of the questions that were hypothesized to result in potential differing responses for land versus oceans. The questions spanned awareness of threats, conservation need, species and system health, conservation intervention type, and funding allocation and source for terrestrial versus marine ecosystems. Responses were either yes or no (coded as 1 or 0, respectively), ranking strongly disagree to strongly agree (1 to 5, normalized 0 to 1 for analysis), none to major need (0 to 4, normalized 0 to 1), very important to not important (1 to 5, normalized 0 to 1), and percent (0–100%) (S1 Table in S1 File).

We evaluated the overall frequency, means, standard deviation, and employed individual Kruskal-Wallis tests, a nonparametric method to determine if the sampled groups (ocean vs. terrestrial) were significantly different (p-value $< 0.05$). All statistical tests were run in R programming v4.3.2 [26], using packages *randomForest* and *ggplot*.

### Random forest analysis

We tested what variables best predict two aspects of conservation: (1) conservation need and (2) perceived environmental impact to self or community. The former gets at the overall feeling of what level of intervention is needed, while the latter centers the personal experience (perceived or real) of environmental impact. In addition, the analysis provides a fuller modeled approach on whether the difference between land and sea are more or less important in

predicting perceived need and impact compared to other variables. The values were normalized from 0 to 1 and were the primary response variables for the Random Forest (RF) models.

We used the highly flexible, machine learning RF analysis to determine explanatory patterns of our two separate response variables of interest (i.e., conservation need and issue impact). RF analysis is a statistical approach that employs recursive and "out-of-bag" bootstrap sampling (i.e., predicting data not in the bootstrap sample) to create binary partitions of predictor variables, fitting regression trees (n = 1000) to the dataset, and finally combining the predictions from all trees [27]. The predictors input into the model are ranked by mean squared error (MSE) [27, 28] and the order reflects the overall influence of each predictor on the response variable in the model. While RF is highly conducive for analyzing diverse social and colinear data, inclusion of multiple predictors with similar levels of MSE typically has diminishing returns on the variance explained (e.g., [29]). As a result, we stepwise pruned the full models and report on the most parsimonious predictive model (i.e., explained approximately equivalent variance of the full model) [30].

We included key attributes described in the literature in predicting conservation need and reported impact to self and/or community. First, we were interested in whether connections to hunting, fishing and/or other recreational activities were more important, and if such trends were indeed generational [3]. Second, we wanted to explore what type of conservation involvement and knowledge (e.g., level of education, awareness of conservation organizations) influenced the strength of responses. Third, we were interested in how political affiliation or connections to the political process (i.e., political affiliation, voting for environmentally supportive politicians, perceived effectiveness of voting) influenced responses around conservation. Lastly, we wanted to get a better sense of how people viewed the funding needs of conservation, if any, specifically where support should come from and how that predicted responses.

A total of 23 questionnaire-based variables were used to model respondent reported conservation need and environmental impact. Eight of the predictor variables were demographic information, including binned age groups (18–24, 25–34, 35–44, 45–54, 54+), gender (male, female), political affiliation (left, right, moderate, won't say), categorical income bracket (low, middle, high, won't say), number of kids, education level (postgraduate, university, vocational technical college, high school, middle school, elementary school, won't say), community type (urban, suburban, rural), and US Census region (Midwest, South, Northeast, West). A 'survey' variable was included to discern any strong terrestrial versus marine division in predicting perceived conservation need or impact. Feelings of concern and perceived habitat or animal health (normalized values 0 to 1) were also included. Familiarity variables included binary (0 or 1) participation in any environmental conservation and awareness (1 or 0) of any conservation organizations or projects. We also included a specific categorical "hunting, fishing, both or neither" variable to determine the strength of the dominance hypothesis [3]. Additionally, we collected information on whether respondents participated in other outdoor activities (snorkeling/diving; boating/sailing; hiking; camping) and summed the total number (0–6). Given funding conservation efforts is a bottleneck, we included several funding predictive variables; one on whether the respondents have donated themselves and three binary (0 or 1) responses on how conservation should be funded: government agencies and/or public-private partnerships, companies whose products damage ocean environments (i.e., regulatory penalties), and private market mechanisms (e.g. payment for ecosystem services). The three funding source variables were selected based on breadth of funding sources and their respective percent of the responses (highest–partnerships and penalties, lowest–market). Regarding politics, and outside of just including political affiliation, we also included binary variables (0 or 1) of whether the respondents voted for environmentally supportive politicians in the last two years

and if they thought voting was effective. Lastly, we included conservation need or impact to self/community (0 to 1) in the opposing models.

# Results

## Demographics

Age, gender, and race were relatively well represented in the study. The 35–44 year olds made up the largest proportion of respondents (31%), followed by 54+ (21%), while the younger (18–24 = 13%) and mid-life respondents (45–55 = 13%) had the fewest people. Gender was relatively equivalent (female 51%), but political affiliation did skew slightly more left (liberal) leaning (37%), than right (conservative; 21%). However, there was a good distribution of the differing affiliations across gender and age (Fig 1). The sample was biased White (70%), with 10% Black, 8% Hispanic or Latino, and 5% Asian, which broadly matches national estimates [31]. It is important to note, the Hispanic or Latino community were under-represented,

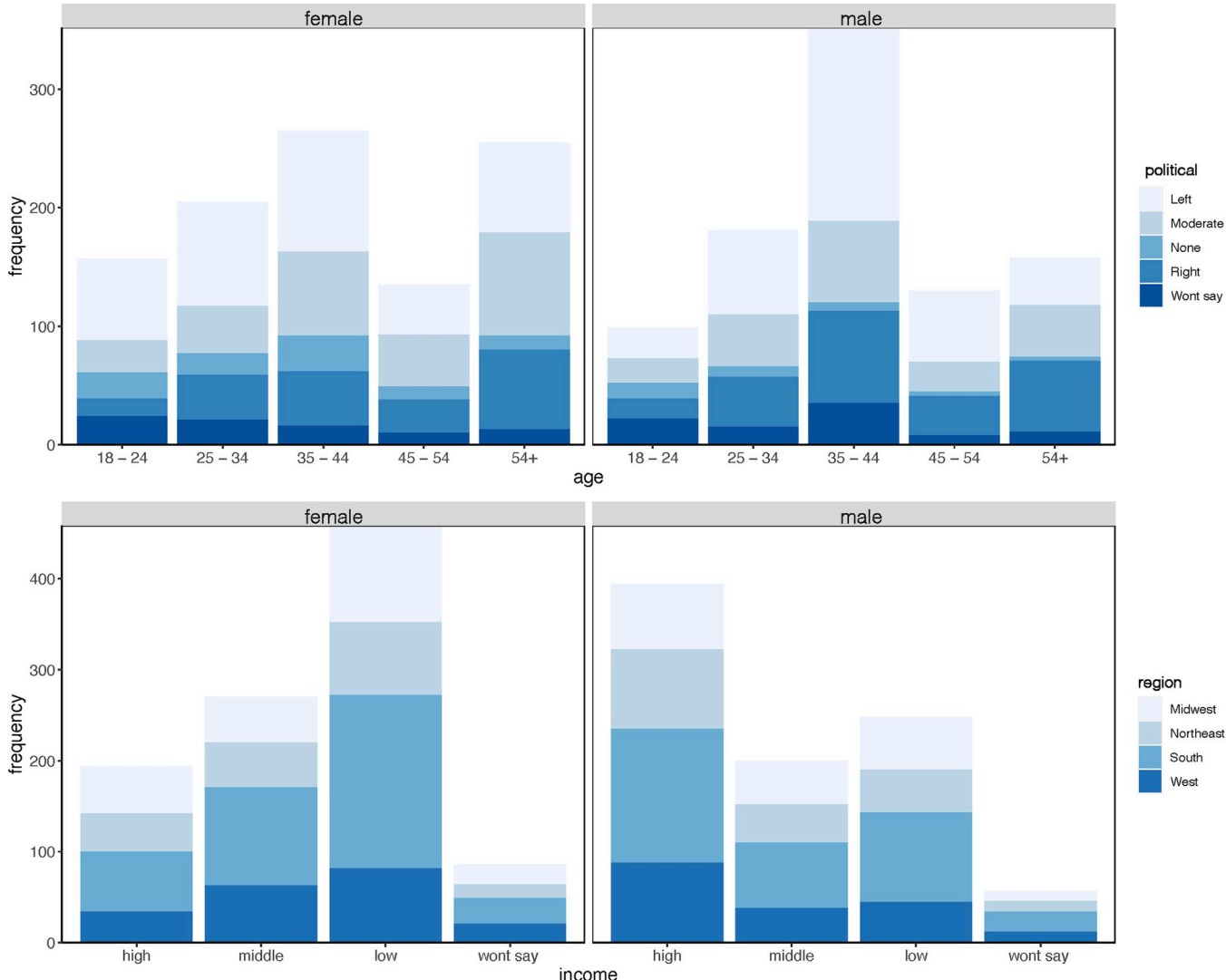

**Fig 1. Demographic distributions of respondents.** Female and male respondents by age (*top panels*) and income (*bottom panels)* colored by political affiliation and U.S. census region.

making up an estimated 19.1% of the actual population in the U.S., as of July 2022. We also recognized races conveyed here are not representative of the diversity of the nation, and are instead a function of PollFish's limited categories.

Social-economically, respondents skewed slightly Southern, were relatively educated, a mix of low to high income, and were more likely involved in fishing than hunting. The South was the most well represented region (37%), followed by the Midwest (19%). We received responses nearly all 50 states (expect Vermont), but Texas, Florida, New York, and California provided the most responses, North and South Dakota the least (S1 Fig in S1 File). Most of the respondents were from cities (47%), followed by suburban (35%) and rural areas (18%). The majority of people received a high school education or higher (87%) and were employed (63%). Interestingly, while gender was relatively even, the income distribution of gender showed a near inverse pattern, with high income males (20%) and low-income females (23%) contributing most to the survey (Fig 1). Lastly, half of the respondents reported hunting and/ or fishing as a recreational activity.

## Terrestrial versus marine

The terrestrial versus marine questions spanned awareness of threats, conservation need, species and system health, conservation intervention effectiveness, and funding allocation and source.

Oceans did garner slightly higher scores around awareness of the issues and coincident lower values concerning the health of the animals or habitats. On average, respondents were slightly, but significantly ($\chi^2$ = 6.4, df = 1, p-value = 0.01) more aware of ocean issues compared to terrestrial (Fig 2A). The trend was largely due to awareness of plastic pollution and oil spills, with 83% and 74% of the ocean survey respondents aware of these issues, respectively. Only extreme temperature/weather came close to plastics or oil at 70% of awareness. However, deforestation (65%)–the number one driver of biodiversity loss globally–did have significantly ($\chi^2$ = 23.5, df = 1, p-value = 1.3e-06) more awareness than overfishing (54%) among

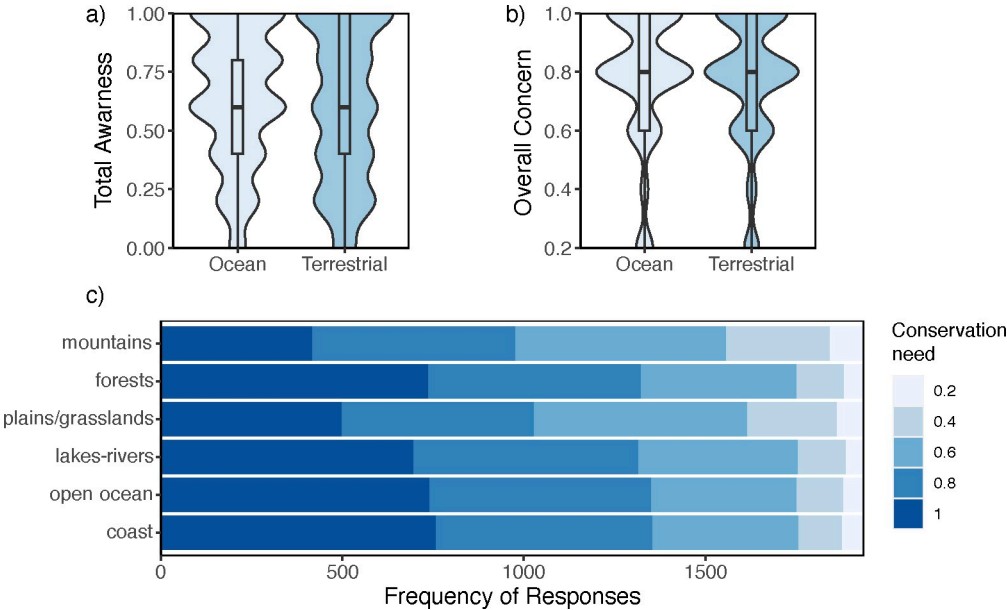

**Fig 2. Ecosystem survey responses.** Collective (a) awareness of issues, (b) overall concern, and (c) conservation need for ocean (*light blue*) versus terrestrial (*dark blue*) habitats and animals.

respondents. In turn, the perceived health of the oceans' animals (mean ± SD = 0.55 ± 0.2; $\chi^2$ = 5.8, df = 1, p-value = 0.016) and habitats (mean ± SD = 0.53 ± 0.2; $\chi^2$ = 16.3, df = 1, p-value = 5.3e-05) were scored significantly lower than terrestrial animals (mean ± SD = 0.57 ± 0.2) and habitats (mean ± SD = 0.57 ± 0.2).

Similar to awareness and perceived health, oceans again garnered slightly higher scores around concern and perceived need. Overall concern was higher on average for oceans, but it was not statistically different ($\chi^2$ = 2.6, df = 1, p-value = 0.11; Fig 2B). However, the view of which ecosystems *need* more conservation were significantly different, with again, aquatic environments–including freshwater–broadly seen as needing more intervention (Fig 2C; $\chi^2$ = 399.21, df = 5, p-value < 2.2e-16). The exception being forests, which received equivalent scoring to marine and freshwater realms.

Ranking of the effectiveness of interventions also slightly favored oceans over terrestrial approaches. When a real-word scenario of oyster restoration versus reforestation were provided, respondents ranked the ocean-based intervention again only slightly higher, but significantly more effective ($\chi^2$ = 9.7, df = 1, p-value = 0.002; S2 Fig in S1 File). Importance of potential conservation approaches of protected areas, captive breeding for reintroductions, relocation, enforcement regulations, environmental education, and supplementing for hunting/fishing were slightly higher for the ocean (mean ± SD across all inventions = 0.83 ± 0.19) than the terrestrial (0.81 ± 0.21) survey, but the overall trends were similar: education, protected areas, and regulations ranking the most important, and supplementing for hunting/fishing the least (S2 Fig in S1 File).

Perceived concern, need, and type of conservation did appear slightly higher for the oceans, but funding allocation and source did not substantially differ. In a hypothetical scenario of how to allocate $1 billion (USD) for environmental conservation programs, respondents in their respective surveys allocated ca. 62% (SD ± 20%) to terrestrial or ocean environments and wildlife ($\chi^2$ = 1.2, df = 1, p-value = 0.27; Fig 3A). This question was specifically designed to account for survey bias of terrestrial versus ocean focus of the questionnaire, showing no real preference for funding for one environment over the other. Where that funding should come from showed similar patterns and the oceans only garnered a slightly higher number of perceived funding sources on average (mean ± SD = 3.2 ± 1.8; $\chi^2$ = 6.2, df = 1, p-value = 0.013) compared to land (3 ± 1.8). More notable was government-linked sources (i.e., penalties, agencies & partnerships, and incentivization by government regulation) were seen more favorably as a source for both systems compared to voluntary pathways, specifically private *market* mechanisms (e.g. payment for ecosystem services) and individual voluntary *donations* to conservation nonprofit organizations (Fig 3B); the exception being companies and corporations with environmental responsibility *programs*.

## Random forest

In both models, conservation need and impact to self and/or community were the best predictors of each other, with slight differences in the top ranked variables. For conservation need, the full model explained 43.4% of the variance [mean of squared residuals (MSR) = 0.017], while the pruned model containing only impact, overall concern for wildlife and habitats, perceived health of habitat, and perceived effectiveness of voting explaining 42.8% of the variance (MSR = 0.017). Notably, impact to self/community explained 30% of the variance. For impact to self and/or my community as the response, the full model captured 46.9% of the variance (MSR = 0.05), and the pruned model with total conservation need, habitat health, penalties on companies as funding source, voted for a pro-environmental candidate in the last 2 years, the total number of recreational activities, and whether they donated to conservation efforts in the last 2 years accounted for 43.0% (MSR = 0.05). Total conservation need explained the vast

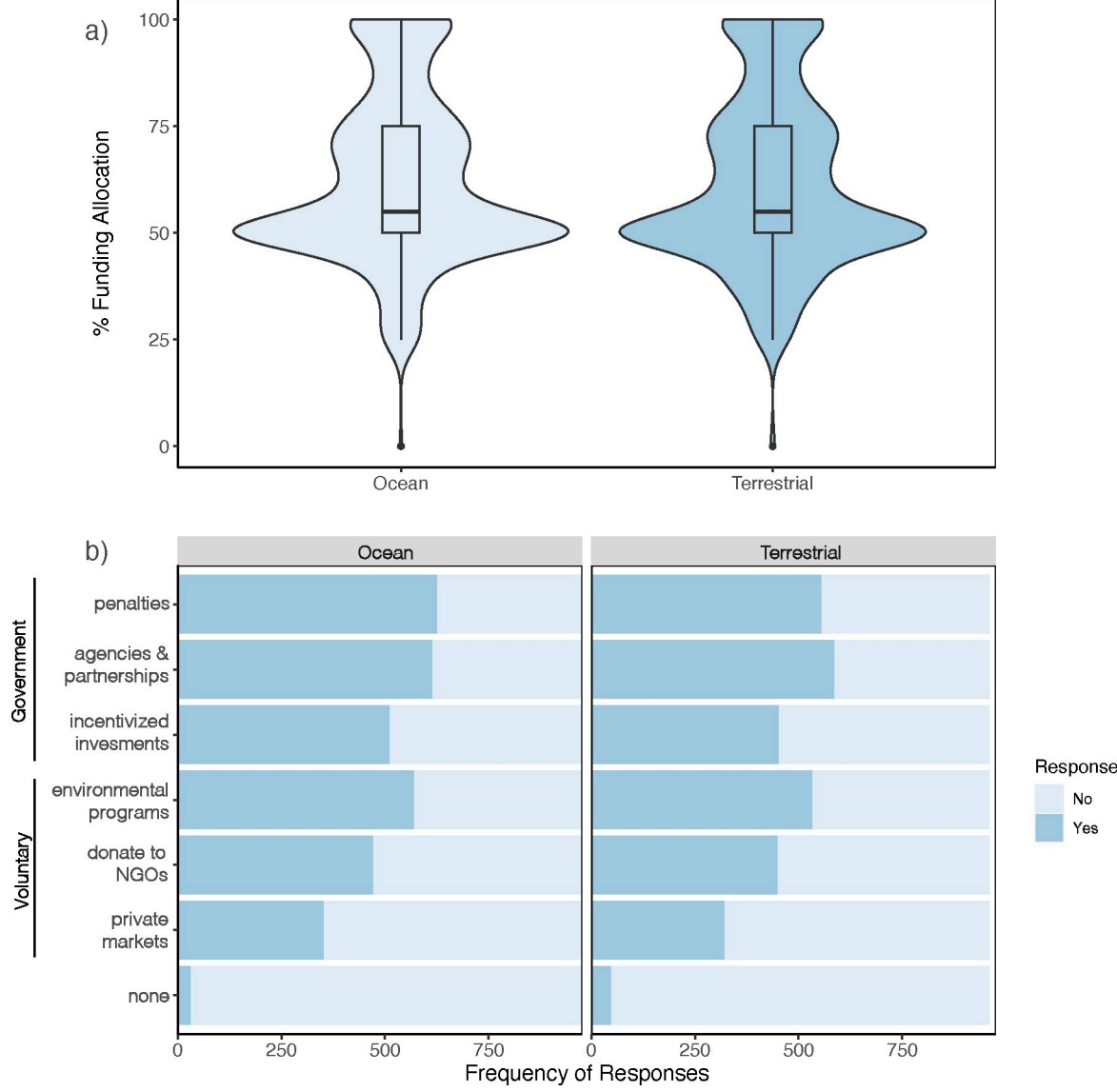

**Fig 3. Funding survey responses.** (a) Hypothetical percent allocation of $1 billion (USD) to ocean (*light blue*) or terrestrial (d*ark blue*) habitats and wildlife conservation from the respective surveys. (b) Response (yes or no) distribution of who should be responsible for funding ocean or terrestrial conservation projects in the U.S. (respondents could select more than one).

majority of variance (29.5%). While additional variables incrementally increased the variance in the impact model, we limited the inclusion of the highly correlated responses (S2 Fig in S1 File) opting for parsimony.

Demographic variables, basic conservation familiarity, and survey (i.e., ocean versus terrestrial) were poor predictors of both models (Fig 4A and 4B). Of the demographic parameters, age and political affiliation ranked comparatively higher, but still only explain 2.8–4.3% of the variance for both models when modeled in isolation. Participation in conservation efforts and knowledge of specific conservation organizations also performed poorly (2.7–3.8%). However, note donation to NGOs or charities in the last 2 years was a moderately important predictor in the impact model (Fig 4B). Most importantly, focus on ocean versus terrestrial systems and species had little to no explanatory power (0–2%) in our models.

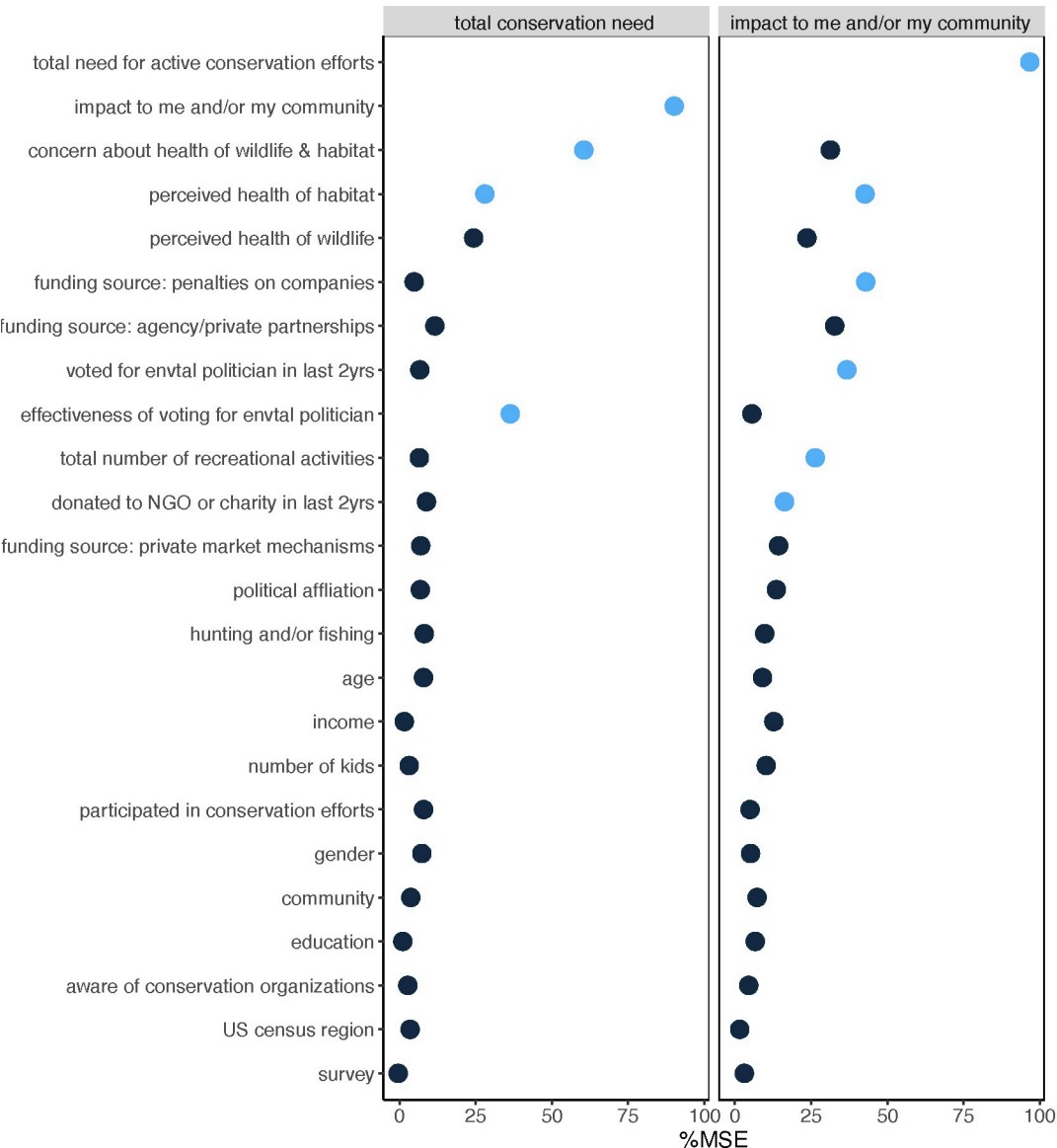

**Fig 4. Random forest model results.** Ranked percent mean standard error (MSE) of predictor variables explaining perceived total conservation need for habitats and wildlife *(left values)* and perceived or real environmental issues impacting me and/or my community *(right panel)*. Predictors for the 'best' pruned models (i.e., near equivalent variance explained as the full model) are depicted in *blue*.

It is notable that the total number of recreational activities ranked highly in the impact model, but hunting and/or fishing less so. Conducting a post-hoc RF partial analysis and descriptive evaluation of hunting/fishing trends, we find a potential threshold of involvement of 2 or more outdoor activities significantly influencing perception of environmental issues impacting them and/or their community (Fig 5A). People who only hunt but do not fish were sparse (4% of response pool: only 28% female and 82% under the age of 45) and reported significantly lower scores than any of the other groups ($\chi^2$ = 30.8, df = 3, p-value = 9.5e-07), including fishing not hunting (Fig 5B; 30% of respondents: 47% female and 67% under 45). Of note, those who reported they hunt *and* fish (17% of respondents: 31% female and 77% under 45) responded equivalently to those who do not engage in either, on average. Moreover,

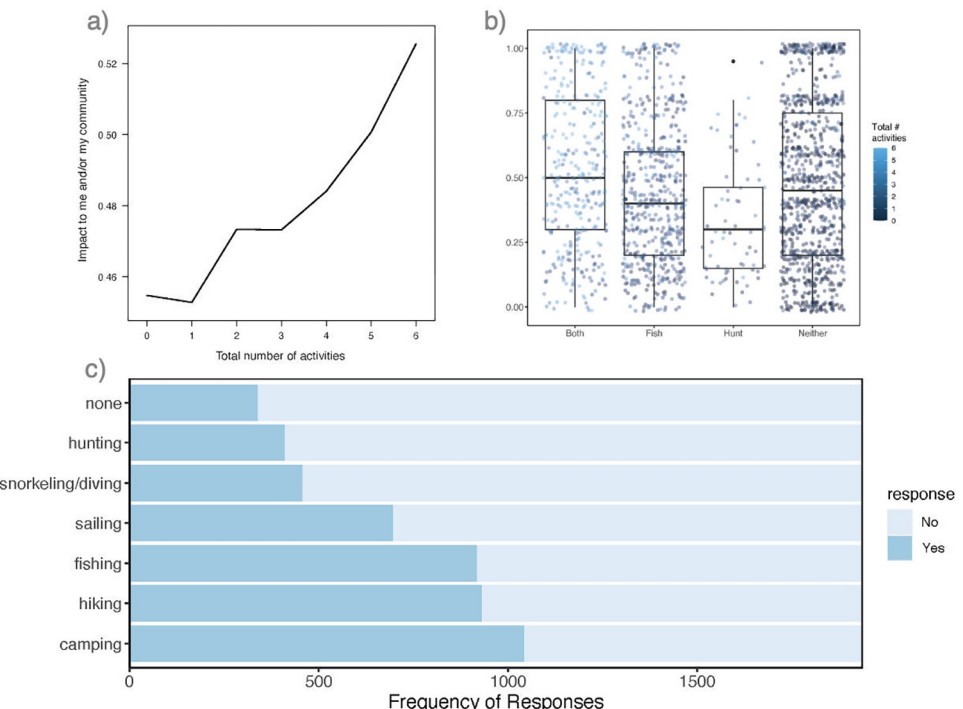

**Fig 5. Outdoor recreational activities.** (a) Partial predictive plot of the total number of outdoor activities on the pruned impact to me and/or my community model and (b) average trends of impact scoring relative to respondents who hunt, fish, both or do neither, as well as the association with other activities (*blue color ramp*); (c) frequency of response of specific outdoor activities.

respondents who hunt and fish also tended to do more average (mean ± SD = 4.6 ± 1.26) outdoor activities in general than the other groups (S3 Fig in S1 File). The most common activities being camping, hiking, and fishing (Fig 5C).

## Discussion

We found evidence of the oceans and marine animals garnering slightly more concern and support than terrestrial systems and organisms, which does not match most global assessments reporting a greater biodiversity crisis on land–especially inland freshwater–than the seas [32–35]. While there is inherent bias associated with ease of measurements on land and lack of coverage of the vast ocean regions, where threats are real and growing [2, 34, 36], the slight significance for ocean ecosystems does raise the question as to the main drivers. While we did not explicitly test for the mechanisms, more detailed awareness is likely an important indicator. Similar to other U.S. specific studies with akin sample sizes [37], plastic pollution and oil spills where the top concerns across both surveys of our study. There have been major social-marketing campaigns around plastic pollution [37, 38], ocean-based oil spills can have very visible and disastrous impacts on animals and local communities [39]–though evidence of singular environmental disasters' influence is mixed [40]–and the ocean may still be seen as more wild, perhaps pristine, compared to the more engrained use and privatization of land [41]. While it is likely a combination of these and other factors, what is driving the slight difference may be worth further investigation. That said, it is important to note the small difference we detected between land and sea did not elicit more (hypothetical) monetary allocation nor predictive power of conservation need.

From a societal perspective, how best to fund the growing conservation need on land and in the oceans was somewhat mixed, but largely reflected how conservation is mostly funded now: government. Respondents tended to favor more government supported pathways for conservation than voluntary mechanisms, so much so that regulatory penalties for companies who pollute–also known as 'polluter pays principle'–was one of the best predictors of perceived impact to respondents and/or their community. Penalties can support conservation in the form of fines/settlements and taxes/fees. Fines and settlements can and have been employed to support conservation efforts, but the monetary influx is variable and the environmental price can be steep. For instance, the 2010 Deepwater Horizon oil spill resulted in the largest environmental settlement in U.S. history ($20.8 billion in 2016) with 80% of those funds going directly to restoration of the impacted region under the RESTORE Act (c. 2012) [42], established in response to the spill. However, it devastated the coast and the associated communities, killing wildlife and decimating oyster reefs and industry in the region [39]. A more sustained source of funding comes from a number of U.S. taxes and fees relevant to biodiversity [43]. Most notably, the Federal Aid in Wildlife Restoration Act (also known as the 1937 Pittman-Robertson Act) taxes the nearly $150 billion spent on hunting and fishing each year [44], contributing ca. $1 billion in a given year in conservation funding [45]. But, hunting participation is decreasing–linked to the decline of 'dominance' values over nature [3]–while funding needs are increasing, suggesting voluntary (market and/or donation) support may be more necessary to diversify resource streams [18, 46]. Indeed, globally conservation funds do not appear to match the estimated need [47]. And although governmental finance is not disappearing but shifting to non-hunting individuals purchasing weapons, the trend is garnering concern around the loss of connection to nature that could result in loss of support for conservation [46].

Recreation, thus access to nature, was an important predictor for the view of environmental issues affecting respondents, something that has been found in many other studies [48–53] and part of the theoretical *Conservation-Recreation Model* [16, 54]. Observational research tends to measure connection to nature as exposure time [48]. We found the number of activities reported by respondents, specifically recreation activities ≥2, was an important predictor of scored impact to self/community, and in turn may influence perceived conservation need—though the direct relationship in this study between outdoor activities and need was weaker. Research has found place-based meaning differs depending on the recreational activity [55], which may influence the direct predictive power of the number of recreational activities on overall conservation need. Nonetheless, people who hunt but did not fish or vice versa did tend to respond with lower overall perceived impact to environmental issues; hunters with the lowest overall scores and sample size, consistent with larger national surveys [44]. However, hunters who also fished had comparable (or higher) values to the general population (i.e., recreating at least 2 activities). Our results provide a more nuanced view on the dominance hypothesis [3]–alongside similar research of hunting-birdwatching [56]–providing additional evidence that it is the diversity of interaction with nature and not necessarily the dominance over nature itself that may be most important in influencing perceived value [17]. Notably, our results support other findings that messaging on a specific issue (e.g., ocean versus terrestrial) is not as important in predicting conservation support, but rather connection to nature in general in multifaceted ways [50]. These findings underscore the need to create easier access to nature in different ways, regardless of the environment; this could be through outdoor programs (e.g., Outdoor Afo, Latino Outdoors, Outward Bound) [57–59], community science engagement (e.g, eBird, COASST) [60–62], and even accessibility apps (e.g., Urban Outdoors), to name a few. That all said, it is also recognized that recreating itself has impact and needs to be managed [63]. Yet the benefits likely outweigh the costs–especially from a human equity

[64] and health perspective [48, 50, 65]–when collective action is needed for protection and management of biodiversity on the whole, land and sea.

Related to recreational connection to nature, we found similar and divergent results compared to other studies, with perceived impact of environmental threats to the respondents themselves and/or their community as key predictors to conservation need, regardless of age. Indeed, research has shown personal experiences or worry (real or perceived) can influence action or interest to addressing climate change [13, 14, 66] and conservation [51–53]. The *Conservation-Recreation Model* posits individual behavior is tightly linked to community, which is informed by the overall connections to nature, as well as social norms [16, 54]https://www.zotero.org/google-docs/?5E65YC. Notably in our study, generation (i.e., age)–a hypothesized important indicator of social differences–was not the most important predictor in either model, which has been found in other research contexts [67] but is somewhat counter to others [3]. However, age did rank on the higher side relative to the other demographic variables, along with income and political affiliation.

Political affiliation was not one of the major predictors of perceived conservation need or environmental issues impacting the respondents, but the political voting process was; a critical component of conservation action [68, 69]. Whether it was the act of voting or the effectiveness of voting for an environmentally supportive politician, voting was one of the most important predictors of how people responded. The distinction in predictive power of political affiliation versus voting could be seen as a glimmer of hope. While there is deep polarization in the U.S., it is how you vote that matters more than perhaps ideology when it comes to conservation of natural ecosystems. In fact, in 2019 a large land conservation policy was re-upped under the Trump administration [70], but attempts to remove ocean protections from fishing was also pursued [71], among others. And although 'number of kids' was not an important predictor of response in our sampled population, connection to nature and family traditions (i.e., personal experiences) can be a tether to conservation, as described above. For example, in 2017, again under the Trump administration, Secretary of the Interior, Ryan Zinke, was quoted in a press release about the U.S. Fish and Wildlife Service recurrent 5-year survey at that time:

> "Some of my best family time growing up and raising my own kids was hunting an elk, enjoying a pheasant, or reeling in a rainbow. These are the memories and traditions I want to share with future generations," [72].

Overall, the importance of conservation is low compared to other perceived issues in the U.S. (e.g., healthcare, the economy), which perhaps makes it less contentious [73] but also less prioritized. These patterns also may not be persistent over time. Other research has found acute changes in perception of conservation before and after elections in a given region [74]. Other researchers have also found ideology is important in driving individual environmental behaviors [75]; the authors did have a much larger sample size during a less polarized period, but did have very different framing in questioning around conservation—in particular linking to "global warming" and trying to answer *why* they conserve. Nonetheless, our results underscore the inherent complexity of social-natural systems.

While our study captured new and re-enforcing patterns related to conservation in the U.S., there were some limitations. We did not include wildlife watching, which makes up a large portion of people and funding in the U.S. (148 million people and total expenditures of $250.2 billion), mostly birding [44]. Most wildlife watching is done at home (99%) and participation is increasing [44], which can be a powerful connection to nature, including diversifying activities [56]. Unlike Manfredo et al. 2021, who showed values toward nature have shifted over

time in the U.S., we did not have a temporal component to the work. However, the intent of this study was largely to test the possible current 'land vs sea' difference, in particular the dominance hypothesis of hunting and/or fishing. Our sample size (N = 1,973) was also comparatively small compared to other somewhat similar studies (e.g., N = 10,000–20,000 [18]), but not others (N = 1,124 [55]), and queried people already interested in responding to a survey (i.e., signed up in PollFish), which could bias the results. For example, our population was more likely to participate in outdoor activities, with ca. 80% involved in at least one of the activities in the study, compared to some national statistics reporting just over half [76]. However, this may be a function of the truncated age range of our study (18 or older). Regardless, the results of this paper should not be taken in isolation, but relative to broader trends in the literature.

In all, we found, like many before us, linking one's self or community to the environment is important for conservation support, but uniquely it does not seem to matter if the connection is terrestrial or ocean-based. Our results are promising, suggesting access and connection to nature more broadly "floats all boats," especially when diversified. Similarly, political affiliation may not be the end all determining conservation support, particularly with stronger connections to nature (e.g., hunting *and* fishing). However, funding remains an issue. While government is the classic and seemingly preferred pathway to support conservation, more voluntary pathways (e.g., private markets) likely need attention for a more mixed economic approach given the growing need—which does seem to be supported by the incoming, voting generation [18]. Ultimately, there is substantial potential to reduce future threats of extinction through increased conservation investments and efforts [34].

## Supporting information

**S1 File. One table and three figures.**
(PDF)

## Acknowledgments

Nearly all data analyses were conducted on Amtrak trains. Given the vital time and connectivity riding trains provided–plus the reduce carbon footprint–we would like to thank California public transit for the indirect support in the completion of this research.

## Author Contributions

**Conceptualization:** Halley E. Froehlich, Darien D. Mizuta, Jono R. Wilson.

**Data curation:** Halley E. Froehlich, Darien D. Mizuta, Jono R. Wilson.

**Formal analysis:** Halley E. Froehlich.

**Funding acquisition:** Halley E. Froehlich, Jono R. Wilson.

**Investigation:** Halley E. Froehlich, Jono R. Wilson.

**Methodology:** Halley E. Froehlich.

**Project administration:** Halley E. Froehlich.

**Resources:** Halley E. Froehlich, Jono R. Wilson.

**Software:** Halley E. Froehlich.

**Supervision:** Halley E. Froehlich.

**Validation:** Halley E. Froehlich, Darien D. Mizuta.

**Visualization:** Halley E. Froehlich.

**Writing – original draft:** Halley E. Froehlich.

**Writing – review & editing:** Halley E. Froehlich, Darien D. Mizuta, Jono R. Wilson.

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
