## [Decision Letter · Decision Letter 0]

14 May 2024

PONE-D-24-10931Public conservation connection and support between ocean and terrestrial systems in the United StatesPLOS ONE

Dear Dr. Froehlich,

Thank you for submitting your manuscript to PLOS ONE. After careful consideration, we feel that it has merit but does not fully meet PLOS ONE’s publication criteria as it currently stands. Therefore, we invite you to submit a revised version of the manuscript that addresses the points raised during the review process.

We look forward to receiving your revised manuscript.

Kind regards,

Amaal Gh. Yasser, Ph.D.

Academic Editor

PLOS ONE

Journal Requirements:

"This project was supported by funding from the The Nature Conservancy (FH02P)."

"This project was supported by funding from the The Nature Conservancy (FH02P)."

"This project was supported by funding from the The Nature Conservancy (FH02P)."

"Jono Wilson is an employee of the Nature Conservancy"

5. In the online submission form, you indicated that your data will be submitted to a repository upon acceptance.  We strongly recommend all authors deposit their data before acceptance, as the process can be lengthy and hold up publication timelines. Please note that, though access restrictions are acceptable now, your entire minimal  dataset will need to be made freely accessible if your manuscript is accepted for publication. This policy applies to all data except where public deposition would breach compliance with the protocol approved by your research ethics board. If you are unable to adhere to our open data policy, please kindly revise your statement to explain your reasoning and we will seek the editor's input on an exemption. 

Reviewers' comments:

Reviewer's Responses to Questions

**Comments to the Author**

1. Is the manuscript technically sound, and do the data support the conclusions?

Reviewer #1: Yes

Reviewer #2: Yes

2. Has the statistical analysis been performed appropriately and rigorously? 

Reviewer #1: Yes

Reviewer #2: Yes

3. Have the authors made all data underlying the findings in their manuscript fully available?

Reviewer #1: Yes

Reviewer #2: Yes

4. Is the manuscript presented in an intelligible fashion and written in standard English?

Reviewer #1: Yes

Reviewer #2: Yes

5. Review Comments to the Author

Reviewer #1: It is suggested to use more results and discussion in the abstract of the article. In the current format, the focus of the abstract is on the introduction

INTRODUCTION

Line 60: You wrote "most US federal policies for conservation are rooted in land-based protections, while ocean-based efforts – mostly passive approaches of marine protected areas...."It is better to mention after this part what reasons they had for paying more attention to the protection of the land than the seas and what factors cause governments and organizations to pay more attention to the protection of the land than the seas.

It is suggested to emphasize the point that different parts of an ecosystem are related to each other and if we consider the earth as a meta-ecosystem, the oceans and land are considered to be part of a single ecosystem, so the emphasis on the mutual effects of land and seas on Each other shows the need to protect both.

Line 70: Your study is based on the dominance hypothesis, so you should first give a brief description of this hypothesis so that it will be easier for the readers of the article to understand the rest of the content.

Line 76: Considering your reference please add some examples for sentence "Education or familiarity with conservation (i.e., conservation literacy) or sustainability activities may play a role in the value of conservation action"

METHODS:

Were the questions used in the questionnaire selected from a specific source?

There are many questions for such studies. What was your criteria for choosing these questions?

Would you please send the codes used for statistical tests in R.

Why did you choose the random forest model for data analysis?

The random forest model suffer from overfitting problem and the output is correlated highly to the input data. How did you over this problem in your analysis?

In the studies that we measure the public opinion and the needs of the society, we must pay attention to the point in which the time and economic conditions of this study are conducted. Have you considered the effect of the time of the survey on the type of response of the society to the questionnaire?

DISCUSSION:

Line 351: Keep in mind that what you have reviewed and studied are the results of questionnaires answered by different groups of people in the community, but what is considered in government and scientific societies for budget allocation and management actions is based on scientific research and projects and precise environmental measurements.

"the oceans and marine animals garnering slightly more concern and support than terrestrial systems and organisms but most global assessments report a greater biodiversity crisis on land" Please explain the reasons of this difference in the views of the public and scientific societies

In general, what is mentioned in the discussion is very similar to the results section of the article, and in fact, in most parts, the results are mentioned without addressing the reasons and interpretation. I recommend rewriting the discussion.

The sample size (N = 1,973) of this study is small, therefore you cannot generalize the results of this study to the entire society, so be careful in choosing words and interpreting the results.

The economic dependence of people on the sea or land ecosystem has a great impact on answering the questions, and this is unfortunately not considered in the interpretation of the results.

Finally, in discussion section, I recommend to use the results of this paper beside broader studies to better comprehend of society view in the case of prioritizing in biodiversity conservation in terrestrial versus ocean

Reviewer #2: Dear Editor/s,

I read the manuscript by Froehlich et al. entitled “Public conservation connection and support between ocean and terrestrial systems in the United States”. The manuscript is highly interesting for me and I guess to the readers. This manuscript provides valuable data. In my idea one of the important parameters that affects views and concerns of the public toward environmental issues in sea of land can be orientation of news and programs produced and broadcasted about these issues. I think it is important to assess effects of social media, news etc. on public attitudes to oceanic or terrestrial environments or biodiversity.

Overall, I think this study is well written which an appropriate set of analyses, which deserve publication in PlosONE.

6. PLOS authors have the option to publish the peer review history of their article (what does this mean?). If published, this will include your full peer review and any attached files.

Reviewer #1: No

Reviewer #2: No

---

## [Author Response · Author response to Decision Letter 0]

21 Jun 2024

PONE-D-24-10931

We would like to thank both reviewers for their time and comments to improve our manuscript. Largely stemming from Review 1, we believe many of the suggestions have substantially improved the paper. In particular, we have improved some of the details pertaining to our approach and expanded upon the drivers that may be behind the slight differences we detected between ocean and terrestrial-based concerns and support. All code and data are also provided via Github, as requested. We aimed to address most of the comments in the manuscript, clarifying those we did not modify, and requesting more detail when a comment was unclear.

Reviewer comments are shown below designated by Q, followed by our responses designated by A:

Reviewer 1:

Q: It is suggested to use more results and discussion in the abstract of the article. In the current format, the focus of the abstract is on the introduction

A:While we certainly want the abstract to convey the motivation and results, we are unsure what the reviewer is requesting we add, as 2/3 of the text touches on the methods and associated findings of the study. If the reviewer could be more specific in their recommendation of what additional aspect should be included or clarified, we are happy to oblige. Below we highlight using [ ]s the portion that is Results and Discussion of the study for easier context:

“Terrestrial and ocean ecosystems are increasingly under threat from an array of anthropogenic pressures. And while threats mount, how people view and value nature is changing. In the United States (U.S.) in particular, there is a shift away from viewing nature as something to ‘dominate,’ as evidenced in the decline in hunting. However, it is unclear if or how opinions around environmental issues and conservation need might differ when comparing ocean versus terrestrial ecosystems, especially given the prevalence and continued importance of wild capture fishing in the U.S. [We employed two national parallel surveys, one focused on oceans, the other land, receiving responses from nearly every state in the U.S. (N = 1,973). While we found only slight, but statistically significant more concern for ocean habitats and animals over terrestrial ecosystems, this did not translate to increased willingness to monetarily support more ocean conservation actions. Using Random Forest models, we also found the best predictor of conservation need was feeling most impacted by environmental issues personally (self and/or community), regardless of ecosystem type. In fact, land versus sea (survey) had the lowest rank in the models, underscoring the importance of general nature-based interactions. Instead, the number of outdoor recreational activities was a highly ranked variable explaining the level of reported impact to self/community, with people who participate in 2 or more activities scoring higher levels of impact, on average. Notably, people who hunt and fish, versus only do one or the other, reported higher levels of impact and participated in more activities overall, providing a more nuanced finding regarding the nature ‘dominance hypothesis.’ Voting, not political affiliation, was also important in explaining responses, and governmental mechanisms to fund conservation were favored over voluntary. Overall, our results add to the strong existing literature that access and connection to nature is key, but uniquely broad connection may “float all boats,” especially when diversified].”

INTRODUCTION

Q: Line 60: You wrote "most US federal policies for conservation are rooted in land-based protections, while ocean-based efforts – mostly passive approaches of marine protected areas...."It is better to mention after this part what reasons they had for paying more attention to the protection of the land than the seas and what factors cause governments and organizations to pay more attention to the protection of the land than the seas.

A:We appreciate this suggestion and have added the following text and additional citations based on the recommendation (line 61-64): “Likely due to a combination of relative accessibility and ease of observation, among other factors, most U.S. federal policies for conservation are rooted in land-based protections, while ocean-based efforts – mostly fishery reforms and passive approaches of marine protected areas (MPAs) – are comparatively recent (Richards, 2018; Hilborn and Ovando 2014).”

Q:It is suggested to emphasize the point that different parts of an ecosystem are related to each other and if we consider the earth as a meta-ecosystem, the oceans and land are considered to be part of a single ecosystem, so the emphasis on the mutual effects of land and seas on Each other shows the need to protect both.

A:We certainly appreciate and recognize the interconnectedness of aquatic and terrestrial ecosystems. However, a systems perspective or approach outside of the sciences is not the typical lens or approach for management. Line 60-76 lays out the argument for key distinctions and treatments. We thus hypothesized there may be a difference in how people perceive the ecosystems, and indeed we did detect slight differences.

Q:Line 70: Your study is based on the dominance hypothesis, so you should first give a brief description of this hypothesis so that it will be easier for the readers of the article to understand the rest of the content.

A:We apologies for the confusion. We do introduce the ‘dominance’ concept in the first paragraph of the Intro (line 55-56), but to make the context clearer we have added the following additional text (lines 73-74): “The ‘dominance hypothesis’ proposes when nature is treated or seen primarily as a resource to exploit for humans there is less consideration for ecosystems.”

Q:Line 76: Considering your reference please add some examples for sentence "Education or familiarity with conservation (i.e., conservation literacy) or sustainability activities may play a role in the value of conservation action"

A:We have added additional context to line 80-81 with discrete examples. The text now reads as follows: “Education or familiarity with conservation (i.e., conservation literacy, such as knowledge of how an ecosystem is impacted) or sustainability activities (e.g., consumer choice) may play a role in the value of conservation action (e.g., Brennan et al., 2019; Jefferson et al., 2021).”

METHODS:

Q:Were the questions used in the questionnaire selected from a specific source? There are many questions for such studies. What was your criteria for choosing these questions?

A: Apologies for the missingness of this important information. While we did not pull exact language from any given study, we did use similar framing from other research. We have added the following text for clarity (lines 122-124): “Questions were informed by the standing literature and developed in consultation with key terrestrial and ocean experts from The Nature Conservancy, as well as specialized survey designers to ensure the questions achieved the intended goal.”

Q:Would you please send the codes used for statistical tests in R.

A:All code and data for stats, models, and figures are now available here: https://github.com/Froehlich-Lab/landsea_usa

Q:Why did you choose the random forest model for data analysis? The random forest model suffer from overfitting problem and the output is correlated highly to the input data. How did you over this problem in your analysis?

A:We chose Random Forest precisely because dealing with diverse and colinear data is one of the strengths of the approach. The method deals with the problem of similar predictors and overfitting by creating multiple trees, with each tree trained slightly differently (via out-of-bag sampling), thus fitting differently for each combination, and then combines the large number (in our case 1,000) decision trees to make the final ranked regression tree. We further deal with ‘overfitting’ by pruning the final tree and removing redundant variables, which are those that have similar ranks and do not greatly reduce the variance explained when removed. 

We explain this in detail in the methods section lines 164-175.

Q:In the studies that we measure the public opinion and the needs of the society, we must pay attention to the point in which the time and economic conditions of this study are conducted. Have you considered the effect of the time of the survey on the type of response of the society to the questionnaire?

A: Time can certainly impact results, which we now more explicitly call out in the Intro (Line 51) and already noted in the in Line 499, in regards political influence. However, to underscore this limitation we have added the following text to the Discussion (line 513-516): “Unlike Manfredo et al. 2021, who showed values toward nature have shifted over time in the U.S., we did not have a temporal component to the work. However, the intent of this study was largely to test the possible current ‘land vs sea’ difference, in particular the dominance hypothesis of hunting and/or fishing.”

DISCUSSION:

Q:Line 351: Keep in mind that what you have reviewed and studied are the results of questionnaires answered by different groups of people in the community, but what is considered in government and scientific societies for budget allocation and management actions is based on scientific research and projects and precise environmental measurements.

A:While we appreciate funding is nuanced, the goal was building off of Larson et al. 2021 on the willingness to voluntarily fund conservation. What we found was less support for those market or donation avenues and more so via government, the current dominant mode. We have updated the text in line 400-405 to better capture this distinction in financial pathways.

We do push back that budget and management is solely informed on sound science and measurements. Social movements and activism, which may be based on science or not, can strongly influence governance structures and actions; climate action is a great example.

Q:"the oceans and marine animals garnering slightly more concern and support than terrestrial systems and organisms but most global assessments report a greater biodiversity crisis on land" Please explain the reasons of this difference in the views of the public and scientific societies

A:Given we found a slight difference in concern, a follow up study to why would be a logical next step. Unfortunately, we can only speculate. That said, we have built out hypothesized reasons into a standalone paragraph and separated out the funding content into a different discussion paragraph (line 361-379). The new text and citations read as follows:

“We found evidence of the oceans and marine animals garnering slightly more concern and support than terrestrial systems and organisms, which does not match most global assessments reporting a greater biodiversity crisis on land – especially inland freshwater – than the seas (Brondizio et al., 2019; Chamberlain, 2020; Isbell et al., 2023; Maxwell et al., 2016). While there is inherent bias associated with ease of measurements on land and lack of coverage of the vast ocean regions, where threats are real and growing (Isbell et al., 2023; McCauley et al., 2015; O’Hara et al., 2019), the slight significance for ocean ecosystems does raise the question as to the main drivers. While we did not explicitly test for the mechanisms, more detailed awareness is likely an important indicator. Similar to other U.S. specific studies with an akin sample size (Baechler et al., 2024), plastic pollution and oil spills where the top concerns across both surveys of our study. There have been major social-marketing campaigns around plastic pollution (Eagle et al., 2016; Baechler et al., 2024), ocean-based oil spills can have very visible and disastrous impacts on animals and local communities (Beyer et al., 2016) – though evidence of singular environmental disasters’ influence is mixed (Farrow et al., 2016) – and the ocean may still be seen as more wild, perhaps pristine, compared to the more engrained use and privatization of land (Rock et al., 2020). While it is likely a combination of these and other factors, what is driving the slight difference may be worth further investigation. That said, it is important to note the small difference detected between land and sea did not elicit more (hypothetical) monetary allocation nor predictive power of conservation need.” 

Q:In general, what is mentioned in the discussion is very similar to the results section of the article, and in fact, in most parts, the results are mentioned without addressing the reasons and interpretation. I recommend rewriting the discussion.

A:We have substantially modified the beginning of the Discussion section and we see now some of the topic sentences could be improved to better reflect the deeper takes and details provided in the associated paragraphs. As such, we have updated several topic sentences and also added additional context throughout for clearer interpretation. If there are more specific aspects our changes have not addressed, we are happy to do so.

Q:The sample size (N = 1,973) of this study is small, therefore you cannot generalize the results of this study to the entire society, so be careful in choosing words and interpreting the results.

A:While we recognize and note our sample size is on the smaller size compared to some studies it is on par with other U.S. wide conservation survey research, which we now note in the main text as well (line 509): 

• Baechler et al. “Public Awareness and Perceptions of Ocean Plastic Pollution and Support for Solutions in the United States.” Frontiers in Marine Science 10 (January 19, 2024): 1323477. https://doi.org/10.3389/fmars.2023.1323477.

o Sample size = 1,960.

• Larson et al. 2018 “Place-Based Pathways to Proenvironmental Behavior: Empirical Evidence for a Conservation–Recreation Model.” Society & Natural Resources 31: 871–91. https://doi.org/10.1080/08941920.2018.1447714.

o Sample size = 1,124

Q:The economic dependence of people on the sea or land ecosystem has a great impact on answering the questions, and this is unfortunately not considered in the interpretation of the results.

A:Fishing and hunting were very logical explicit connectors to aquatic vs land activities that we did explore and discuss in detail. Economic dependence is quite tricky, especially relative to land. If the reviewer has a specific dimension they would like us to include in the Discussion we are happy to explore their recommendation if more detail is provided.

Q:Finally, in discussion section, I recommend to use the results of this paper beside broader studies to better comprehend of society view in the case of prioritizing in biodiversity conservation in terrestrial versus ocean

A:We cite ca. 50 studies and reports in the Discussion section, comparing and linking to our results. With the slightly new structure, spurred by the reviewers above comments, we address the following main points in each Discussion paragraph as follows:

1st paragraph: Reconciling the difference in concern and need for oceans, highlighting marketing campaigns, environmental disasters, and broader perceived ‘wildness’ of the ocean as potential drivers.

2nd paragraph: We note how to fund the growing need was largely related to how we fund it now, via government, discussing how that is done and changing. In particular, decline in hunting, which may require voluntary pathways to fill the gap.

3rd paragraph: We highlight the extensive and long history of identifying recreation/access to nature as key to conservation, e.g., Conservation-Recreation Model, while also discussing the nuance around the ‘dominance hypothesis’ we found. In addition, we provide concrete examples from the literature of how to increase connection.

4th paragraph: We touch upon personal experience influencing behavior, as well as age not being an important predictor which matches some but not other studies.

5th paragraph: We discuss political affiliation versus voting, again comparing our findings to personal values and findings from other studies.

6th paragraph: Our caveats paragraph, e.g., we did not include bird watching like a similar study, again comparing to othe

---

## [Editor Report · Decision Letter 1]

5 Jul 2024

Public conservation connection and support between ocean and terrestrial systems in the United States

PONE-D-24-10931R1

Dear Dr. Halley,

We’re pleased to inform you that your manuscript has been judged scientifically suitable for publication and will be formally accepted for publication once it meets all outstanding technical requirements.

Kind regards,

Amaal Gh. Yasser, Ph.D.

Academic Editor

PLOS ONE

---

## [Editor Report · Acceptance letter]

16 Jul 2024

PONE-D-24-10931R1 

PLOS ONE

Dear Dr. Froehlich, 

I'm pleased to inform you that your manuscript has been deemed suitable for publication in PLOS ONE. Congratulations! Your manuscript is now being handed over to our production team.

Kind regards, 

on behalf of

Dr. Amaal Gh. Yasser 

Academic Editor

PLOS ONE